# Preparation of $Ce_x$-$Mn_{0.8}Fe_{0.2}O_2$ Catalysts and Its Anti-Sulfur Denitration Performance

**Yelin Zhang [1], Chao Zhang [1], Yusi Wang [1], Li Zhang [1], Jing Zeng [1] and Hanbing He [1,2,*]**

[1] School of Metallurgy and Environment, Central South University, Changsha 410083, China
[2] Key Laboratory of Metallurgical Emission Reduction & Resources Recycling, Anhui University of Technology, Ma'anshan 243002, China
[*] Correspondence: hehanbinghhb@163.com; Tel.: +86-13875985605

**Abstract:** In order to meet the industrial denitrification demands, inexpensive ferrous metals Mn and Fe have been chosen as the raw materials for the catalysts of CO-SCR, and the anti-sulfur denitrification performance of ferromanganese catalysts can be greatly enhanced by Ce doping. In this study, $Ce_x$-$Mn_{0.8}Fe_{0.2}O_2$ catalysts were prepared by co-precipitation, and the effects of Ce addition on the structure and morphology of prepared catalysts and their anti-sulfur denitration performance were investigated with X-ray diffraction (XRD), scanning electron microscopy (SEM) and X-ray photoelectron spectroscopy (XPS). The results showed that the $Ce_x$-$Mn_{0.8}Fe_{0.2}O_2$ catalysts consisted of nanoparticles sized 20–100 nm. Specifically, the $Ce_{0.2}$-$Mn_{0.8}Fe_{0.2}O_2$ catalyst had more active sites and the best anti-sulfur denitration performance, with a denitration rate of 90.36% at 350 °C, while the denitrification performance of the $Mn_{0.8}Fe_{0.2}O_2$ catalyst was only 85%. Furthermore, the denitrification rate of the catalyst was maintained above 80% when the $CO:NO:SO_2$ ratio was 3:1:1 for 4 h at 350 °C.

**Keywords:** ferromanganese catalyst; Ce doping; anti-sulfur denitration; CO-SCR

## 1. Introduction

Nitrogen oxides ($NO_x$) and sulfur dioxide ($SO_2$) emissions from industrial flue gas are primary air pollutants [1,2]. The most commonly used catalyst in the industry is $V_2O_5$ + $WO_3(MoO_3)/TiO_2$ [3,4], which suffers from a narrow catalyst activity temperature range, a poor thermal stability, and the ability to easily react with $SO_2$ to form ammonia sulfate salt when ammonia is used as the reducing agent. In addition, when CO is used as the reducing agent instead of $NH_3$, CO, $NO_x$ and $SO_2$ can interact well with each other to realize simultaneous desulfurization and denitrification, as well as solve the problem of CO pollution [5]. Therefore, exploring catalysts with low cost, good thermal stability, many active sites, and strong sulfur resistance for industrial denitrification processes is the current research focus in the denitrification field.

Manganese oxide catalysts with multivalency, strong redox capability, and good performance in denitrification have been a hot research topic in recent years [6,7]. Metal doping is an effective method used to improve the denitrification performance and sulfur resistance of catalysts [8,9]. For example, the addition of iron oxide can not only promote the uniform dispersion of active components but also improve the NO conversion [10]. Zhao [11] prepared an Fe-Mn/AC catalyst via the impregnation method. The denitrification efficiency of the obtained Fe-Mn/AC catalyst could reach 90% at 120~200 °C and 70% with the presence of $SO_2$. Tian [12] prepared ferromanganese catalysts with the co-precipitation and spray-drying method. The catalysts are porous microsphere structures that can provide larger specific surface areas and more active sites for the adsorption and activation of the reactant gases, and their denitrification rate can reach 90% at 250 °C. Since the active components such as $Fe_2O_3$ and $Mn_2O_3$ are susceptible to sulfation, they are usually modified by bimetallic doping to improve their sulfur resistance [13,14].

CeO$_2$ is widely used in catalytic denitrification because of its good redox properties, good oxygen storage and release capacity, and easy conversion between Ce$^{4+}$ and Ce$^{3+}$. Furthermore, the addition of Ce to manganese-based composite metal oxides can effectively inhibit sulfate generation and improve sulfur resistance [15]. Yang [16] prepared Mn-Ce/biomass charcoal (BC) catalysts by loading Mn and Ce onto BC using the impregnation method. The denitrification rate of the catalyst could be maintained at 80% under an SO$_2$ concentration of 200 ppm at 170 °C. During the reaction, oxygen transfers from CeO$_2$ to Mn$_2$O$_3$ to promote the cyclic catalytic reaction rate and therefore significantly improve the NO conversion in the MnCe/BC catalyst. In addition, Fang [17] used density functional theory to study the relationship between Ce doping on catalyst denitration reaction and SO$_2$ oxidation, and they found that NO and SO$_2$ were more easily adsorbed on the catalyst surface after Ce doping and could increase the reaction difficulty of SO$_2$ oxidation to SO$_3$. Zhang [18] investigated the effects of Mn and Ce loading sequences and molar ratios on denitrification and sulfur resistance using fly ash as a catalyst carrier. The results showed that when the Mn, Ce bimetal was simultaneously loaded, the manganese ions entered the cerium oxide lattice and formed a solid solution with an Mn-O-Ce fluorite structure, which gave the catalyst the best denitrification and sulfur resistance performance. The co-doping of Ce in the catalyst can not only reduce the reaction temperature of the catalytic reaction but also effectively inhibit the sulfidation of the catalyst, stabilize the denitrification performance, and improve the service life of the catalyst.

Therefore, in this study, the inexpensive metal elements Mn and Fe were selected as the main components of the catalysts, and Ce was used to modify the ferromanganese catalyst. The structure and denitration performance of the catalysts for selective catalytic reduction were investigated.

## 2. Results and Discussion

### 2.1. CO-SCR and Anti-Sulfur Performance

Figure 1 shows the denitrification rates of the MF catalysts with different doping amounts of Ce. The denitrification performance of the Ce$_x$-MF catalyst increased with the increase in Ce when the Ce doping was less than 0.4. When the Ce doping amount was 0.2, the denitrification rate reached 90% at 350 °C. Compared with the MF catalyst, the Ce$_{0.2}$-MF catalyst had the best denitrification performance. The high Ce content may have resulted in the formation of Ce oxides that could cover or block the active sites, leading to a decrease in the denitrification rate [15–17].

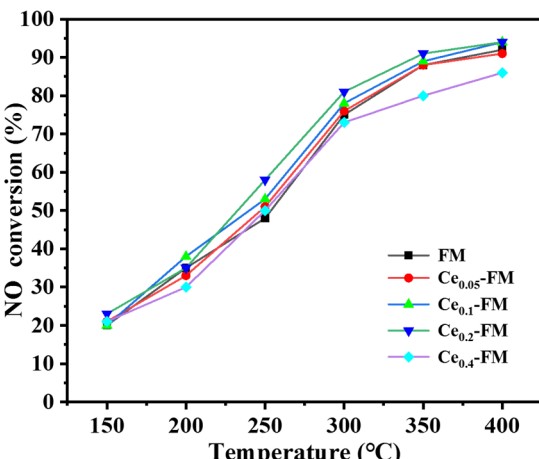

**Figure 1.** Effect of Ce doping ratio on the denitration performance of the catalysts.

Figure 2 shows the denitrification rate curves of the MF and Ce$_{0.2}$-MF catalysts with different SO$_2$ concentrations at 350 °C. For the MF catalyst, the denitration rate was significantly reduced after the introduction of SO$_2$, and the denitration rate of the catalyst

could not return to the original level after the introduction of $SO_2$ was stopped. For the $Ce_{0.2}$-MF catalyst, when the $SO_2$ concentration was 200 ppm, the denitration rate slightly decreased after the introduction of $SO_2$, and the denitration rate returned to the original level after stopping the introduction of $SO_2$. The incorporation of Ce significantly enhanced the anti-sulfur denitrification of the MF catalyst when the $SO_2$ concentration was lower than 400 ppm. However, when the $SO_2$ concentration further increased, the denitrification rate of the catalyst significantly decreased, demonstrating that the catalyst after Ce doping was still not suitable at a high sulfur concentration.

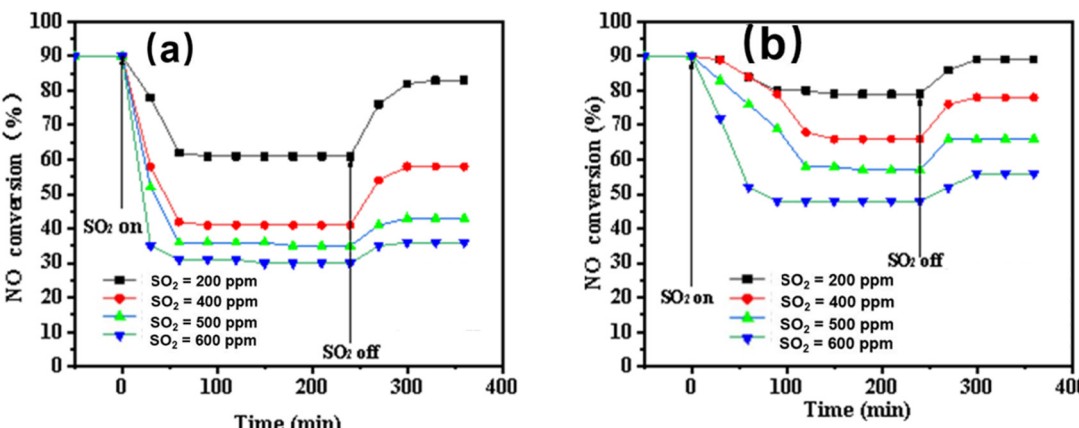

**Figure 2.** (**a**) Effect of different $SO_2$ concentrations on the denitration activity of the MF catalyst; (**b**) effect of different $SO_2$ concentrations on the denitration activity of the $Ce_{0.2}$-MF catalyst.

### 2.2. Characterization of $Ce_x$-MF Catalysts

Figure 3 shows the XRD patterns of the $Ce_x$-MF catalysts with different doping amounts of Ce. The diffraction peaks are attributed to the $Mn_2O_3$(PDF Standard Card #41-1442) phase [19]. The intensity of the diffraction peaks weakened and moved towards higher angles as the content of Ce increased. No characteristic peak of Fe oxide can be observed in the figure, indicating that Fe was well-dispersed in the catalyst. As the content of Ce increased to 0.2 and 0.4, the diffraction peaks of $CeO_2$ emerged, demonstrating that a small amount of Ce was present in the form of $CeO_2$. A small amount of Ce doping provided oxygen to $MnO_x$, therefore improving the denitrification efficiency of the catalyst. The synergistic effect between $CeO_2$ and $Mn_2O_3$ also enhanced the denitration performance of the $Ce_{0.2}$-MF catalyst.

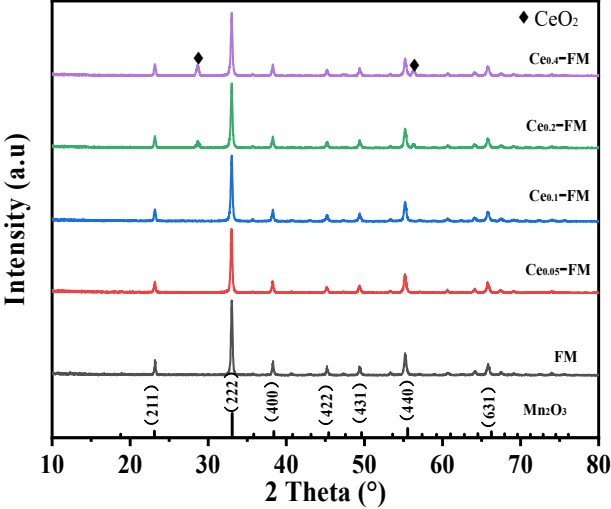

**Figure 3.** XRD patterns of $Ce_x$-MF catalysts.

The specific surface area, pore volume, and average pore size of the $Ce_x$-MF catalysts were analyzed, and the results are shown in Table 1. The specific surface area of the $Ce_x$-MF catalysts ranged from 54 to 77 nm and rapidly increased with the increase in Ce doping content, but the pore capacity and pore size showed decreasing trends. The $Ce_{0.4}$-MF catalyst was prepared with the largest specific surface area of 77.543 $m^2$ $g^{-1}$ and an average pore size of 3.824 nm. Combined with the CO-SCR activity and BET results, there was a correlation between specific surface area and catalytic activity. The CO-SCR reaction mainly took place on the surface of the catalyst, which was conducive to the adsorption and activation of the reaction, thus increasing the denitration activity of the catalyst. The higher specific surface area, the more active sites that are available. These results confirmed that although the specific surface area affects the denitrification performance of catalysts, it is not the most important factor in a denitrification reaction; rather, the active components have synergistic effects on each other.

**Table 1.** Pore structure characteristics of the catalysts with different Ce doping proportions.

| Sample | Specific Surface Area ($m^2$ $g^{-1}$) | Pore Volume ($cm^3$ $g^{-1}$) | Average Pore Diameter (nm) |
|---|---|---|---|
| MF | 49.702 | 0.208 | 10.387 |
| $Ce_{0.05}$-MF | 54.565 | 0.197 | 9.585 |
| $Ce_{0.1}$-MF | 61.169 | 0.189 | 7.778 |
| $Ce_{0.2}$-MF | 68.489 | 0.165 | 7.125 |
| $Ce_{0.4}$-MF | 77.543 | 0.115 | 3.824 |

The CO-TPR profiles of the catalysts are shown in Figure 4. The $Ce_x$-MF catalysts had an obvious reduction peak at 350~500 °C. The intensity of the reduction peak tended to increase as x increased, and the intensity and area of the reduction peak reached a maximum at x = 0.2. Overall, the $Ce_{0.2}$-MF catalyst showed a stronger catalytic reduction ability.

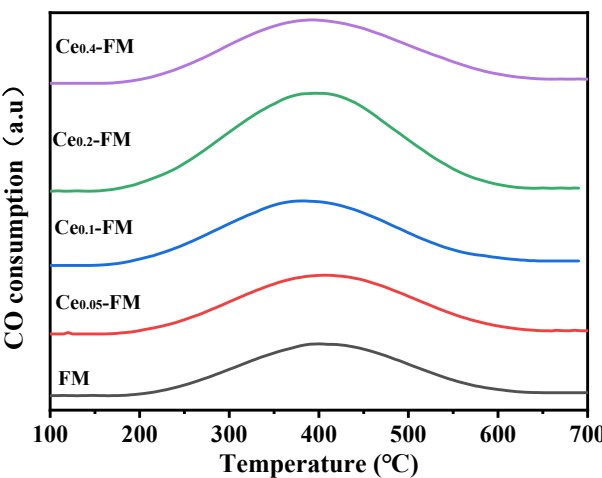

**Figure 4.** CO-TPR profiles of $Ce_x$-MF catalysts.

Figure 5 shows the surface morphology of the catalysts with different Ce doping proportions. The $Ce_x$-MF catalysts had an irregular shape, with a particle size ranging from 20 to 100 nm. The as-prepared catalysts were fluffy and porous, which facilitated the flow of gas and allowed for a larger contact area between the catalyst and the reaction gas, as well as providing more active sites on the surface, thus improving the denitration reaction. The $Ce_{0.2}$-MF catalyst particles were more uniform in size, while the $Ce_{0.4}$-MF catalyst particles were uneven in size and had more obvious agglomeration, indicating that a small amount of Ce doping was improved the dispersion of the elements and the denitrification performance of the catalyst. Furthermore, the X-ray energy spectrum elemental surface

distribution analysis (SEM-EDAX) of the $Ce_{0.2}$-MF sample is displayed in Figure 6, showing that Mn, Fe, Ce and O were uniformly dispersed in the catalysts.

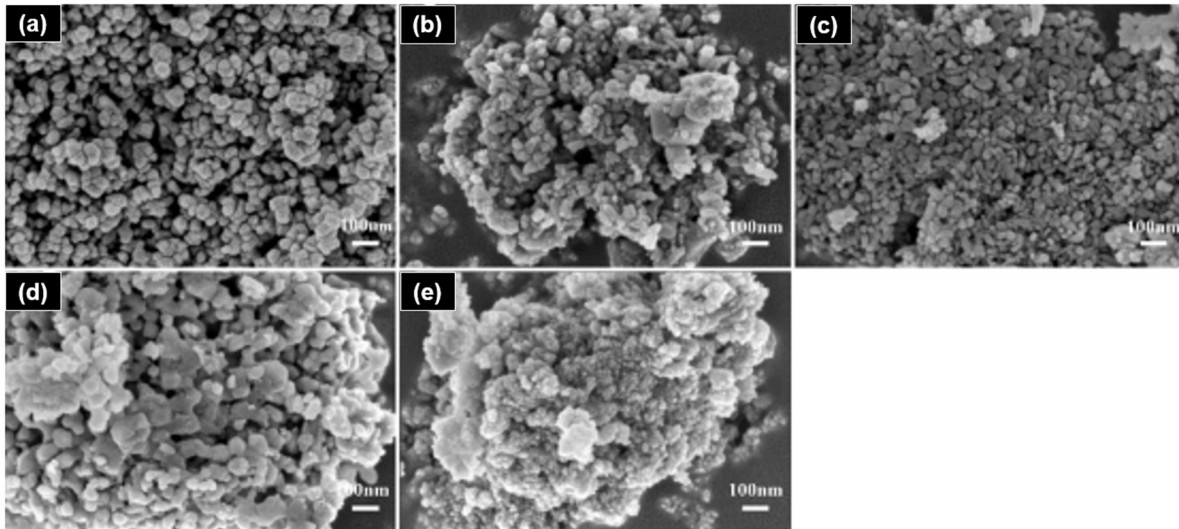

**Figure 5.** SEM images of the catalysts: (**a**) MF, (**b**) $Ce_{0.05}$-MF, (**c**) $Ce_{0.1}$-MF, (**d**) $Ce_{0.2}$-MF, and (**e**) $Ce_{0.4}$-MF.

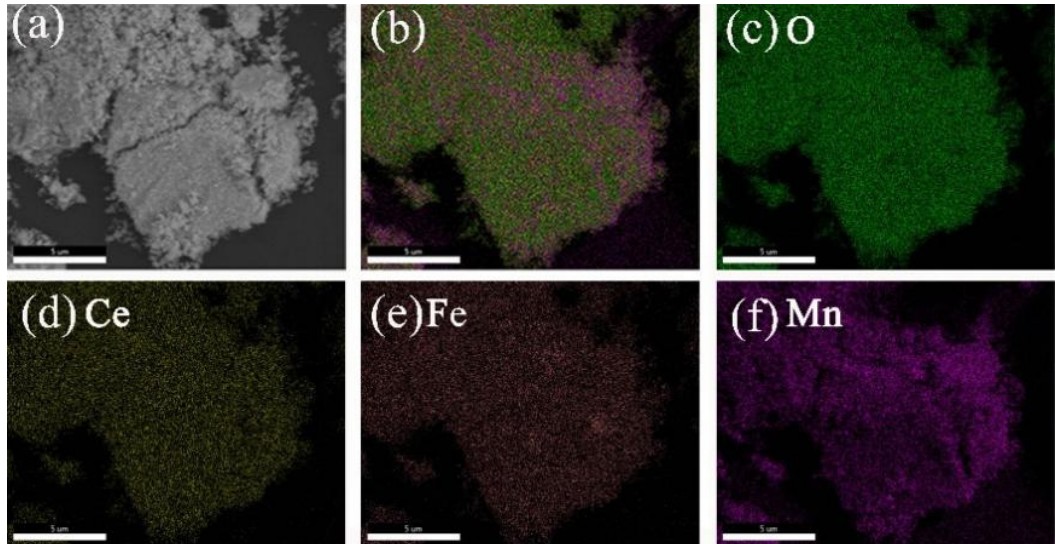

**Figure 6.** SEM image and EDAX element mapping distributions of $Ce_{0.2}$-MF: (**a**) SEM image; (**b**) element overlay: (**c**) O, (**d**) Ce, (**e**) Fe, and (**f**) Mn.

In order to investigate the effect of Ce incorporation on the anti-sulfur denitrification performance of the MF catalysts, XPS analyses of the MF and $Ce_{0.2}$-MF catalysts were carried out (Figures 7a–d and 8a,b). Figure 7a shows the Mn2p spectra of the MF catalyst before and after the sulfur resistance. Mn2p3/2 appeared at approximately 641.4 eV before the anti-sulfur reaction and could be divided into two peaks: 641.2 eV attributed to $Mn^{3+}$ and 642.5 eV attributed to $Mn^{4+}$ [20]. After the anti-sulfur reaction, Mn2p3/2 was divided into three peaks, 640.1 eV attributed to $Mn^{2+}$, 641.2 eV attributed to $Mn^{3+}$, and 642.5 eV attributed to $Mn^{4+}$. Figure 8a shows the Mn2p spectra of the $Ce_{0.2}$-MF catalyst before and after the sulfur resistance. Mn2p3/2 appeared at approximately 641.4 eV before the anti-sulfur reaction, similar to that of the MF catalyst. The incorporation of Ce increased the $Mn^{4+}$ content of the catalyst. In addition, the catalyst did not contain $Mn^{2+}$ after the anti-sulfur reaction, revealing that the incorporation of Ce effectively inhibited the production

of $MnSO_4$. Furthermore, the $Mn^{4+}$ content in the $Ce_{0.2}$-MF catalyst decreased and the $Mn^{3+}$ content increased, as shown in Table 2, indicating that Mn participated in the redox reaction process in the reaction process. Therefore, the conversion of $Mn^{4+}$ to $Mn^{3+}$ was the control reaction in the entire reaction process. $Mn^{4+}$ is more catalytically active than $Mn^{3+}$, so the reduced $Mn^{4+}$ content after the anti-sulfur reaction also corresponded to a reduction in denitrification performance.

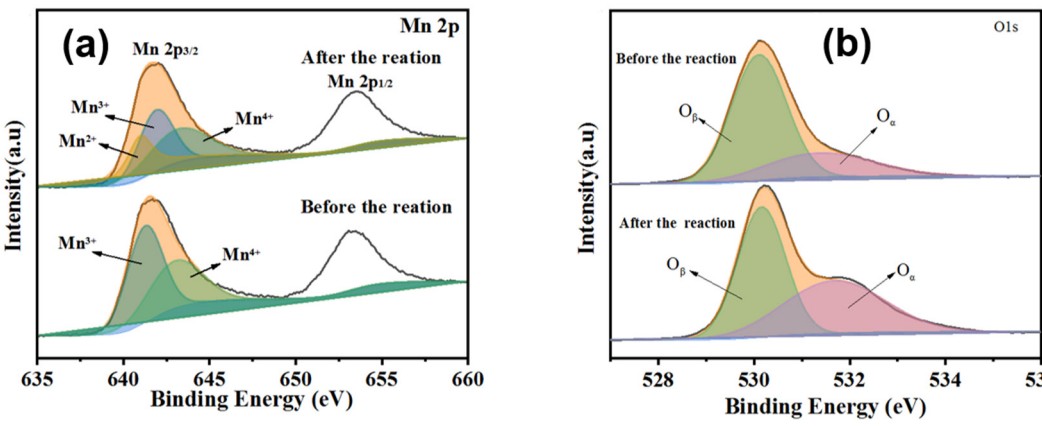

**Figure 7.** The XPS spectra of the MF catalyst before and after the anti-sulfur reaction: (**a**) Mn2p; (**b**) O1s.

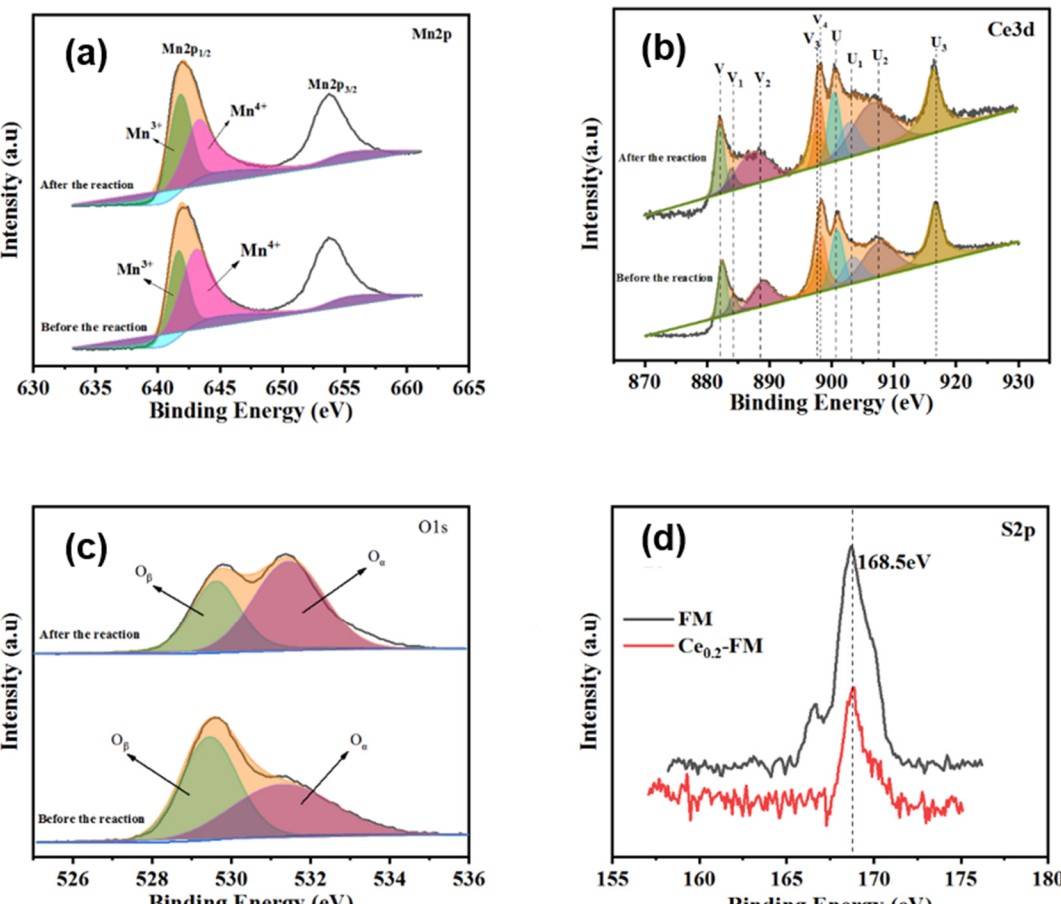

**Figure 8.** The XPS spectra of $Ce_{0.2}$-MF catalyst before and after the anti-sulfur reaction: (**a**) Mn2p; (**b**) Ce3d; (**c**) O1s; (**d**) S2p spectra of the $Ce_{0.2}$-MF and MF catalysts after the anti-sulfur reaction.

**Table 2.** XPS analysis results of MF and $Ce_{0.2}$-MF catalyst before and after the anti-sulfur reaction.

| Sample | Mass Fraction/% | | | | | | |
|---|---|---|---|---|---|---|---|
| | $Mn^{2+}$ | $Mn^{3+}$ | $Mn^{4+}$ | $Ce^{3+}$ | $Ce^{4+}$ | $O_\beta$ | $O_\alpha$ |
| MF catalyst | / | 56.5 | 44.5 | / | / | 62.4 | 38.6 |
| MF catalyst after anti-sulfur reaction | 27.9 | 36.8 | 35.3 | / | / | 53.4 | 46.6 |
| $Ce_{0.2}$-MF catalyst | / | 48.6 | 51.4 | 24.4 | 75.6 | 66.3 | 33.7 |
| $Ce_{0.2}$-MF catalyst after anti-sulfur reaction | / | 56.7 | 43.3 | 30.9 | 69.1 | 48.6 | 51.4 |

The peaks of the Ce3d spectra in the $Ce_{0.2}$-MF catalyst could be divided into nine peak types (as shown in Figure 8b), where V (882.5 eV), V2 (888.8 eV), V3 (898.4 eV), U (901.0 eV), U2 (907.5 eV), and U3 (916.7 eV) are attributed to $Ce^{4+}$; V1 (884.9 eV), V4 (898.8 eV) and U1 (903.5 eV) are attributed to $Ce^{3+}$ [21]; and the proportion of $Ce^{4+}$ was 75.6% and the proportion of $Ce^{3+}$ was 24.4%. The ratio of $Ce^{4+}$ to $Ce^{3+}$ in the catalytic reaction was an important parameter used to measure the performance of the catalysts for denitrification. The higher the ratio, the easier to gain electrons during the reaction and the more oxygen vacancies that can be formed in the catalyst. Therefore, the interconversion between $Ce^{3+}$ and $Ce^{4+}$ will provide the catalyst with the corresponding oxygen vacancies, thus improving the denitrification performance of the catalyst [22]. Table 2 shows that the $Ce^{4+}$ content decreased after the reaction and that $SO_2$ preferentially reacted with the Ce to $Ce_2(SO_4)_3$, resulting in an increase in $Ce^{3+}$ content, though only a small amount of metal sulphate was generated after the reaction.

The O1s spectra before and after the MF and $Ce_{0.2}$-MF catalyst anti-sulfur reaction were divided into two peaks (as shown in Figures 7b and 8c) that were related to the redox nature of the metal ions. The peak with the lower electron-binding energy (528–529 eV) belonged to the lattice oxygen peak, labeled $O_\beta$, while the peak with the higher electron-binding energy (531–532 eV) belonged to the chemisorbed oxygen peak [23], labeled $O_\alpha$. The $O_\beta$ provided oxygen vacancies to the catalyst and plays an important role in the improvement of denitrification performance, and the $O_\alpha$ was mainly present on the catalyst surface and interacted with the $SO_2$ adsorbed on the catalyst surface to form sulphate during the reaction. As can be seen from Table 2, oxygen in the $Ce_{0.2}$-MF catalyst mainly existed in the form of $O_\beta$, and the $O_\beta$ content decreased as the oxygen vacancies were gradually occupied during the anti-sulfur reaction while the denitrification rate of the catalyst decreased, indicating that the presence of $O_\beta$ in this reaction system greatly improved the denitrification performance of the catalyst. It can be seen from Table 2 that Ce doping can enhance the $O_\beta$ content of the catalysts to a certain extent [24], providing oxygen vacancies for the catalysts and thus facilitating the catalytic reaction.

Figure 8d shows the S2p orbitals of the catalyst after the anti-sulfur reaction. It can be seen from the figure that the characteristic peak of the S2p orbital mainly appeared at 168.5 eV, which corresponds to $SO_4^{2-}$, and its corresponding peak intensity and peak area represent the amount of $SO_4^{2-}$ produced during the reaction. The characteristic peak of S2p orbital of $Ce_{0.2}$-MF catalyst was smaller than that of the MF catalyst in both peak intensity and peak area. Scholars have found that the addition of Ce can generate a Ce-O-Mn solid solution on the surface of a catalyst, and the formation of the solid solution can enhance the Lewis acidity of the catalyst, inhibit the adsorption of sulfur dioxide on its surface, and reduce the loss of Lewis acid sites by inhibiting the adsorption of sulfur dioxide. Moreover, the addition of Ce also weakens the sulfate stability on the catalyst surface, thus reducing the deposition of sulfate on the catalyst surface [25].

In order to investigate the surface change of catalysts with the presence of $SO_2$, ex situ SEM before and after the denitration reaction was also carried out. As shown in Figures 5a and 9a, the surface of the MF catalyst was covered with a white substance, which was $CeSO_4$ after the anti-sulfur reaction according to Figure 7a. However, Figure 9b shows that a small amount of white material appeared on the surface of the $Ce_{0.2}$-MF

catalyst after the anti-sulfur reaction, and the main component of this white material was $Ce_2(SO_4)_3$ according to Figure 8b.

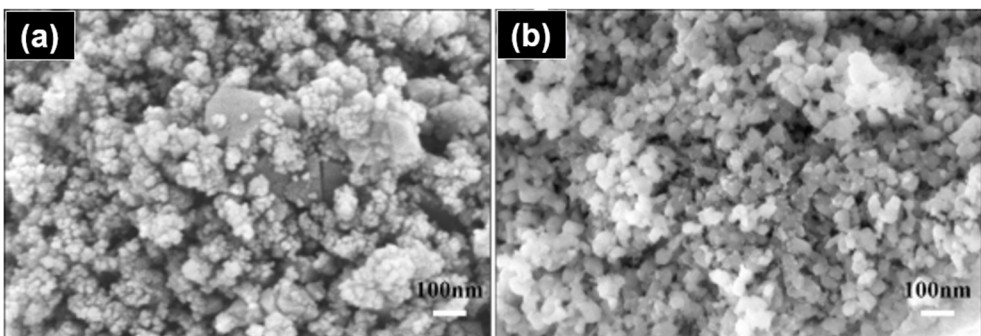

**Figure 9.** SEM image of the catalyst after the anti-sulfur reaction: (**a**) MF and (**b**) $Ce_{0.2}$-MF.

## 3. Experimental

### 3.1. Preparation of Catalysts

The catalysts had the best denitrification performance when the Mn:Fe molar ratio was 8:2 according to our previous study [26]. The specific preparation process of the catalyst was as follows: 1.96 g of $CH_3(COO)_3Mn·4H_2O$ (Sinopharm Chemical Reagent Co., Ltd., Shanghai, China), 0.81 g of $Fe(NO_3)_3·9H_2O$ (Sinopharm Chemical Reagent Co., Ltd., Shanghai, China), and x g of $Ce(NO_3)_3·6H_2O$ (x = 0.21, 0.43, 0.86, and 1.73 g) (Sinopharm Chemical Reagent Co. LTD, China) were placed in 40 mL of ultrapure water and stirred thoroughly for 1 h. Then, $NH_3·H_2O$ (25 wt.%) (Sinopharm Chemical Reagent Co., Ltd., Shanghai, China) was added dropwise to control the end point as pH = 10. After stirring for another 5 h, the precipitation was washed by distilled water several times and then dried at 80 °C for 10 h. The obtained solid was ground into powder and placed into a muffle furnace (Sinopharm Chemical Reagent Co., Ltd., Shanghai, China) with a heating rate of 5 °C/min to 600 °C for 5 h. The $Ce_x$-$Mn_{0.8}Fe_{0.2}O_2$ catalysts were denoted as $Ce_x$-MF (x = 0.05, 0.1, 0.2, and 0.4, where x is the molar ratio of Ce to Mn).

### 3.2. Catalyst Structure and Morphology Characterization

A JSM-6360LV-type electron scanning microscope (SEM, Nippon Electronics Co, Tokyo, Japan) was used to observe the surface microscopic morphology and analyze the elemental composition of the materials via energy spectral surface scanning. An X-ray diffractometer (XRD) from Rigaku D, Tokyo, Japan was used to examine the physical phase structure of the synthesized material with Cu target ($\lambda$ = 1.5406 Å), a working voltage of 40.0 KV, a scanning range 2θ of 10–80°, and a scanning speed of 10 °/min. Additionally, Thermo Fisher's ESCALABXi+ (Kandel, Germany) was used to determine the electronic structure of the solid surface and the chemical composition of the surface composition.

### 3.3. Characterization of Catalyst Denitration Performance

First, 0.5 g of the prepared catalysts was placed into a tube furnace. $N_2$ with 500 ppm of NO and 500 ppm of CO was used as the reaction gas at a flow rate of 200 mL $min^{-1}$. The reaction was carried out at a rate of 5 °C/min to 400 °C. For the anti-$SO_2$ denitration, CO and $SO_2$ were introduced at a volume ratio of CO:$SO_2$ = 2:1 ($SO_2$ was taken as 200, 400, 500 and 600 ppm) after the sample denitrification rate was stable. The CO-TPR experiments comprised 0.5 g of catalyst in a tubular reaction furnace with a 1% CO and 99% $N_2$ gas mixture, a total gas flow rate of 200 mL $min^{-1}$ and a ramp-up rate of 5 °C $min^{-1}$ so that the reaction temperature was uniformly increased to 700 °C for the ramp-up reduction reaction, changing the reaction temperature and catalyst type for the experiment.

The denitrification rate η of NO concentration at the outlet was detected and recorded in real time using a flue gas analyzer, and (η) was calculated as follows:

$$\eta = \frac{\alpha - \beta}{\alpha} \times 100\% \tag{1}$$

where $\alpha$ is the NO concentration at the inlet and $\beta$ is the $NO_x$ concentration at the outlet, including NO and $NO_2$

## 4. Conclusions

The $Ce_x$-MF catalysts with different Ce contents were prepared with a co-precipitation method. The obtained catalysts were irregularly shaped nanoparticles with a particle size of 20–100 nm. Ce existed in the form of $Ce^{3+}$ and $Ce^{4+}$ in the Ce-MF catalyst. The doping of small amount of Ce helped to refine the particles, increase the content of $Mn^{4+}$ and lattice oxygen, and effectively inhibit sulfate generation and improve its sulfur resistance, resulting in a better anti-sulfur denitration performance. The denitrification rate of the $Ce_{0.2}$-MF catalyst was above 80% after 4 h with an $SO_2$ concentration of 200 ppm, which was 20% higher than that of the MF catalyst. This simple and inexpensive method will boost the industrial application of Mn-Fe catalysts.

**Author Contributions:** Data curation, Y.Z.; formal analysis, Y.Z., C.Z., Y.W. and L.Z.; resources, H.H.; supervision, J.Z. and H.H.; writing—original draft, Y.Z.; writing—review & editing, Y.Z. All authors have read and agreed to the published version of the manuscript.

**Funding:** National Key R&D Program of China (Grant No. 2021YFB3701400) is greatly appreciated.

**Data Availability Statement:** Not applicable.

**Conflicts of Interest:** The authors declare that they have no known competing financial interest or personal relationships that could have appeared to influence the work reported in this paper.

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
