# Peer review of "Preparation of Cex-Mn0.8Fe0.2O2 Catalysts and Its Anti-Sulfur Denitration Performance"

_catalysts, doi:10.3390/catal12101141_

Round 1

Reviewer 1 Report

The article presents clear and direct language that facilitates the review, however some specific issues need to be better discussed or presented.

Author Response

Many thanks for your comments concerning our manuscript entitled “Preparation of Cex-Mn0.8Fe0.2O2 catalysts and its anti-sulfur denitration performance”. The main corrections and the responses to reviewer’s comments are given as follows:

Comments :

*In item 3.1. CO-CSR performance

  1. a) in lines 112-113...the authors must insert explanations or justifications related to the possible effects of the ionic states of Ce on the conversion of NO, as supported by the XPS results.
  2. b) In general, it is necessary to present other results from the literature that compare the performance of the Ce0.2 catalyst against other inorganic materials with activity for COSCR.

**In item 3.2. Characterization of Ce-MF catalysts

  1. c) In line 160, a discussion is recommended for the observed fact that the catalyst Ce0.4 has a lower profile for CO-TPR than the analogue Ce0.2.

Response:

Thank you for the comment. We will answer them all below:

*a). We added references to explain it.

 b). We think that the key point of this paper is to compare the sell-out ability of different contents of Ce ion doped manganese matrix composite oxide, and the main comparison object is ourselves.

**We revised the article and expounded it directly.

Reviewer 2 Report

The authors' studies need to be supplemented with experimental results regarding the selectivity of the synthesized catalysts compared to the selectivity of conventional Fe- and Mn-based catalysts. For gas denitrification, a very important feature is the selectivity of the catalysts towards the NO to N2 reduction reaction.
Also, the authors must specify the gas space velocity used.
The preparation and characterization of the catalysts are well structured and argued.
Based on the catalysts characterizations, the authors can suggest a reaction mechanism scheme, through which to highlight the influence of carbon and sulfur oxides over NOx catalytic destruction.

The authors must specify in the summary and conclusions the innovative aspects of the catalytic process in which the synthesized catalysts are used.

Author Response

Many thanks for your comments concerning our manuscript entitled “Preparation of Cex-Mn0.8Fe0.2O2 catalysts and its anti-sulfur denitration performance”. The main corrections and the responses to reviewer’s comments are given as follows:

Comments :

The authors' studies need to be supplemented with experimental results regarding the selectivity of the synthesized catalysts compared to the selectivity of conventional Fe- and Mn-based catalysts. For gas denitrification, a very important feature is the selectivity of the catalysts towards the NO to N2 reduction reaction.

Also, the authors must specify the gas space velocity used.

The preparation and characterization of the catalysts are well structured and argued.

Based on the catalysts characterizations, the authors can suggest a reaction mechanism scheme, through which to highlight the influence of carbon and sulfur oxides over NOx catalytic destruction.

The authors must specify in the summary and conclusions the innovative aspects of the catalytic process in which the synthesized catalysts are used.

Response:

Thank you for the comment. Compared with traditional materials, the synthetic material in this paper has improved the gas storage and release ability and sulfur resistance ability by inserting Ce. And the gas velocity was 200mL/min, it was difficult to calculate gas space velocity due to irregular samples and containers. And we have added to the summary and conclusions to indicate innovative aspects of the catalytic process.